# The genetic characterization of fall armyworm populations in Ecuador and its implications to migration and pest management in the northern regions of South America

Rodney N. Nagoshi[1]*, Ernesto Cañarte[2], Bernardo Navarrete[2], Jimmy Pico[2], Catalina Bravo[2], Myriam Arias de López[3], Sandra Garcés-Carrera[2]

1 Center for Medical, Agricultural and Veterinary Entomology, United States Department of Agriculture-Agricultural Research Service, Gainesville, Florida, United States of America, 2 National Institute of Agriculture Research (INIAP), Quito, Ecuador, 3 Research Consultant, Guayaquil, Ecuador

* rodney.nagoshi@ars.usda.gov

**Data Availability Statement:** Relevant DNA sequence data are available in GenBank (accession

## Abstract

The fall armyworm (*Spodoptera frugiperda*) is a moth pest native to the Western Hemisphere that has recently become a global problem, invading Africa, Asia, and Australia. The species has a broad host range, long-distance migration capability, and a propensity for the generation of pesticide resistance traits that make it a formidable invasive threat and a difficult pest to control. While fall armyworm migration has been extensively studied in North America, where annual migrations of thousands of kilometers are the norm, migration patterns in South America are less understood. As a first step to address this issue we have been genetically characterizing fall armyworm populations in Ecuador, a country in the northern portion of South America that has not been extensively surveyed for this pest. These studies confirm and extend past findings indicating similarities in the fall armyworm populations from Ecuador, Trinidad-Tobago, Peru, and Bolivia that suggest substantial migratory interactions. Specifically, we found that populations throughout Ecuador are genetically homogeneous, indicating that the Andes mountain range is not a long-term barrier to fall armyworm migration. Quantification of genetic variation in an intron sequence describe patterns of similarity between fall armyworm from different locations in South America with implications for how migration might be occurring. In addition, we unexpectedly found these observations only apply to one subset of fall armyworm (the C-strain), as the other group (R-strain) was not present in Ecuador. The results suggest differences in migration behavior between fall armyworm groups in South America that appear to be related to differences in host plant preferences.

## Introduction

The Noctuidae family include a number of moth species that cause substantial agricultural damage. One extensively studied example is *Spodoptera frugiperda* (J. E. Smith) (Lepidoptera:

numbers provided in manuscript). All other relevant data are included in the manuscript.

**Funding:** The author R.N.N. received support came from the Agricultural Research Service of the United States Department of Agriculture (6036-2200-30-00D) and USAID PASA (908-0210-012). The authors E.C., B.N., J.P., C.B., M.A., and S.G.C. received support from the National Institute of Agriculture Research (INIAP), Ecuador, authorized by the agreement between the INIAP and the Ministry of Environment of Ecuador reference number MAE-DNB-CM-2015-0024.

**Competing interests:** The authors have declared that no competing interests exist.

Noctuidae), commonly called fall armyworm. This species is native to the Western Hemisphere but has recently become a global agricultural pest [1–3]. Although permanent populations are limited to tropical and subtropical climates, fall armyworm is capable of rapid and extensive migrations and exhibits high genetic diversity. The latter manifests itself in a broad host range (reports of associations with over 80 plant species) and a high incidence of pesticide resistance [4–7]. These traits make fall armyworm an invasive threat with the potential to be difficult to control.

Economic damage by fall armyworm occurs primarily in corn and sorghum with more sporadic but significant infestations reported in rice, millet, pasture and forage grasses, cotton, and sugarcane [8–10]. It is the primary insect pest of corn in the southeastern United States, the Caribbean, and South America [11]. This broad host range is in part due to the presence of two subpopulations that differ in their host plant preferences [12, 13]. Historically designated as host strains, their phylogenetic relationship remains unclear in large part because they are for all practical purposes morphologically indistinguishable. There is a suggestion of wing size differences between strains in South America [14, 15], but this appears to be an environmental consequence of the plant host used during development [16]. A number of physiological and behavioral differences have been reported between the strains [13, 17–24]. However, such studies are complicated by the high genetic variability and regional differences exhibited by the species, which can confound the demonstration of consistent strain differences between different laboratory colonies [25].

The strains are most consistently identified using genetic markers from portions of the mitochondrial *Cytochrome Oxidase Subunit I* (*COI*) and the Z-chromosome-linked *Triosephosphate isomerase* (*Tpi*) genes [26, 27]. One segment of *COI*, COIB, carries polymorphisms instrumental in distinguishing the strains as well as two geographically distinct groups designated as the FL-type and TX-type [28, 29]. The nuclear *Tpi* gene appears to be a more accurate indicator of strain identity [27] and the use of more variable intron sequences potentially increases the resolving power of this marker for distinguishing subgroups [30]. Both *COI* and *Tpi* encode for housekeeping functions that are not believed to contribute to strain differences in behavior. The association of *Tpi* with strain identity suggests a physical linkage with the one or more functions driving strain divergence and thereby a sex chromosome-based inheritance pattern for strain identity.

The strains were originally designated corn-strain and rice-strain for the host plants they were isolated from [13]. However, subsequent studies demonstrated a wider host range and more variable host-specificity, so we now refer to the two groups as the C-strain (preferentially found in corn, sorghum, and cotton) and R-strain (rice, millet, pasture and forage grasses). A generally consistent correspondence between the strain markers and preferred host plants have been demonstrated for fall armyworm populations in North America and South America, though the association is not absolute. For example, on average about 80% of the fall armyworm larvae collected from corn express the expected C-strain haplotypes (*COI*-CS and TpiC) with the remainder showing the R-strain markers (*COI*-RS and TpiR) [8, 9, 31]. In addition, while the *COI* and *Tpi* markers are generally in agreement with respect to strain identity, substantial disagreements are sometimes observed depending on collections [8, 10, 31] and have turned out to be the norm in populations newly discovered in the Eastern Hemisphere [32–35]. One likely cause of this discordance between markers is hybridization between strains, which has been demonstrated in both the laboratory and in field populations though at lower frequencies than mating within strains [27, 32, 36, 37]. Given these observations we assume that current methods can only estimate the most likely strain identity of an individual specimen and should be assumed to approximate the strain composition of a population.

Long distance migration has been documented for fall armyworm in North America. Permanent populations winter in two locations separated by the Gulf of Mexico, southern Texas/Mexico and southern Florida [4]. Annual northward migrations from these locations begin in the spring in conjunction with favorable seasonal winds and progressive northward planting of corn, the latter providing large acreages capable of supporting high density fall armyworm populations [38]. Fall armyworm is capable of sustained flights of up to 12 hours during nocturnal hours at high altitudes [38]. These environmental and biological conditions when modeled are sufficient to explain the several thousand kilometers migration observed over the growing season thereby identifying wind patterns and host availability as the major determinants of migration behavior [39].

Similar long distance migration of fall armyworm in other regions has not been demonstrated though it is suspected to have contributed to the rapid dissemination of the pest in the Eastern Hemisphere [40, 41] and rapid spread of resistance traits between the Caribbean and South American populations [6, 42, 43]. Genome-level studies suggest a mostly single breeding population in the Western Hemisphere, consistent with substantial migration [44], though there are indications of regional genetic heterogeneity in South America [45]. Previous studies have also identified statistically significant differences in haplotype frequencies between C-strain populations that winter in Texas (TX-type) and Florida (FL-type) that have persisted for over a decade of observations [46, 47]. Taken together, these studies indicate substantial gene flow throughout the hemisphere, but also identify regions where interactions between populations are insufficient to produce genetic homogeneity.

An important question relevant to mitigating the spread of pesticide resistance traits is the degree to which permanent populations throughout the Western Hemisphere intermix. To partly address this issue, we recently reported a study of fall armyworm from cornfields in the Manabi province of Ecuador that by location is a potential contributor to fall armyworm in Central America and the Caribbean [48]. The collection site was predominated by "hard corn", a variety typically used by the feed industry. The collection displayed the *COI* markers indicative of the TX-type, similar to that found in Mexico and the rest of South America, but differing from most of the Caribbean and Florida populations. We now extend the survey to include multiple locations in Ecuador to examine the level of homogeneity of the fall armyworm in the country, the impact of the Andes mountain range on migration, and to test for the existence of the two strains. This includes the first examination of infestations in "soft corn" varieties, a traditional staple subsistence crop in the region. Collections from soft corn, hard corn, and rice were compared for their haplotype profiles using the *COI* and *Tpi* markers with the collection sites representing a cross-section of Ecuador. These were also compared to populations from other South American locations. The implications of these findings on the migration of fall armyworm in the Western Hemisphere are discussed.

## Materials and methods

### Specimen collections and DNA preparation

Ecuador specimens were obtained in 2018–2019 as larvae from soft corn and hard corn hosts from multiple provinces and from rice plants in Guayas, Loja, and Manabi provinces (Table 1, Fig 1A). Soft corn is typically grown in the highlands region and is used for food while hard corn is mostly found in tropical low elevations and used as animal feed. Collections were made at sites with high levels of cultivation of corn and rice that provide multiple sampling locations for the three major ecological regions in Ecuador, the coastal lowlands (Coast), the Andes highlands (Andes), and the Amazon tropical forest (Amazon). Collections were made between February 2018 and October 2019, with times dictated by the sowing and harvest dates that

**Table 1. Source information for fall armyworm collections.** Number refers to map locations in Fig 1. Asterisk indicates collection from [48].

| Site | Province | Date | Host | Region | Coordinates |
|------|----------|------|------|--------|-------------|
| a | Azuay | Mar-May 2018 | soft corn | Andes | -2.798 -78.767 |
| b | Bolivar | Mar 2018 | soft corn | Andes | -1.713 -79.034 |
| c1 | Imbabura | Jul 2018 | soft corn | Andes | 0.427 -78.185 |
| c2 | Imbabura | Dec 2018 | soft corn | Andes | 0.303 -78.271 |
| c3 | Imbabura | Dec 2018 | soft corn | Andes | 0.238 -78.255 |
| d1 | Loja | Mar 2018 | hard corn | Andes | -3.878 -79.645 |
| d2 | Loja | Mar 2018 | hard corn | Andes | -4.067 -79.851 |
| d3 | Loja | Mar 2018 | hard corn | Andes | -4.116 -80.107 |
| e* | Manabi | Mar 2018 | hard corn | Coastal | -1.056 -80.425 |
| f1 | Morona-Santiago | Aug 2018 | hard corn | Amazon | -2.618 -78.209 |
| f2 | Morona-Santiago | Aug 2018 | hard corn | Amazon | -2.308 -78.120 |
| g | Napo | Aug 2018 | hard corn | Amazon | -0.995 -77.816 |
| h1 | Sucumbios | Aug 2018 | hard corn | Amazon | -0.068 -76.882 |
| h2 | Sucumbios | Aug 2018 | hard corn | Amazon | -0.189 -76.642 |
| i1 | Guayas | Oct 2019 | rice | Coastal | -2.678 -79.617 |
| i2 | Guayas | Oct 2019 | rice | Coastal | -2.095 -79.694 |
| j1 | Loja | Aug 2018 | rice | Andes | -4.380 -79.939 |
| j2 | Loja | Aug 2018 | rice | Andes | -4.386 -80.242 |
| k | Manabi | Jul 2019 | rice | Coastal | -0.922 -80.449 |

varied by location and crop. Typically, fall armyworm collections were of larvae at approximately stage L3-4 (~10–14 days post-hatching) obtained 2–3 weeks after corn germination and about 5 days after rice germination. Collections described previously and used in this study include those from hard corn in Manabi province (2018, [48]) and a mixture of feed corn and sweet corn from Peru (2014, [28]), Trinidad and Tobago (2013, [28]), Bolivia (2012, [28]), Argentina (2011, [8]), and Brazil (2005, [29]) (Fig 1B). No endangered or protected species were involved in this study. Permission was obtained from private farmers for access and data collection in their fields.

Collected specimens were stored either air-dried or in ethanol at room temperature or refrigerated. A portion of each specimen was excised and homogenized in a 5-ml Dounce homogenizer (Thermo Fisher Scientific, Waltham, MA, USA) in 800 µl Genomic Lysis buffer (Zymo Research, Orange, CA, USA) and incubated at 55°C for 5–30 min. Debris was removed by centrifugation at 10,000 rpm for 5 min. The supernatant was transferred to a Zymo-Spin III column (Zymo Research, Orange, CA, USA) and processed according to manufacturer's instructions. The DNA preparation was increased to a final volume of 100 µl with distilled water. Genomic DNA preparations of fall armyworm samples from previous studies were stored at -20°C. Species identity was initially determined by morphology and confirmed by sequence analysis of the COIB region.

## PCR amplification and DNA sequencing

PCR amplification for all segments was performed in a 30-µl reaction mix containing 3 µl 10X manufacturer's reaction buffer, 1 µl 10mM dNTP, 0.5 µl 20-µM primer mix, 1 µl DNA template (between 0.05–0.5 µg), 0.5-unit Taq DNA polymerase (New England Biolabs, Beverly, MA). The thermocycling program was 94°C (1 min), followed by 28 cycles of 92°C (30 s), 56°C (45 s), 72°C (45 s), and a final segment of 72°C for 3 min. Typically 96 PCR amplifications were performed at the same time using either 0.2-ml tube strips or 96 well microtiter

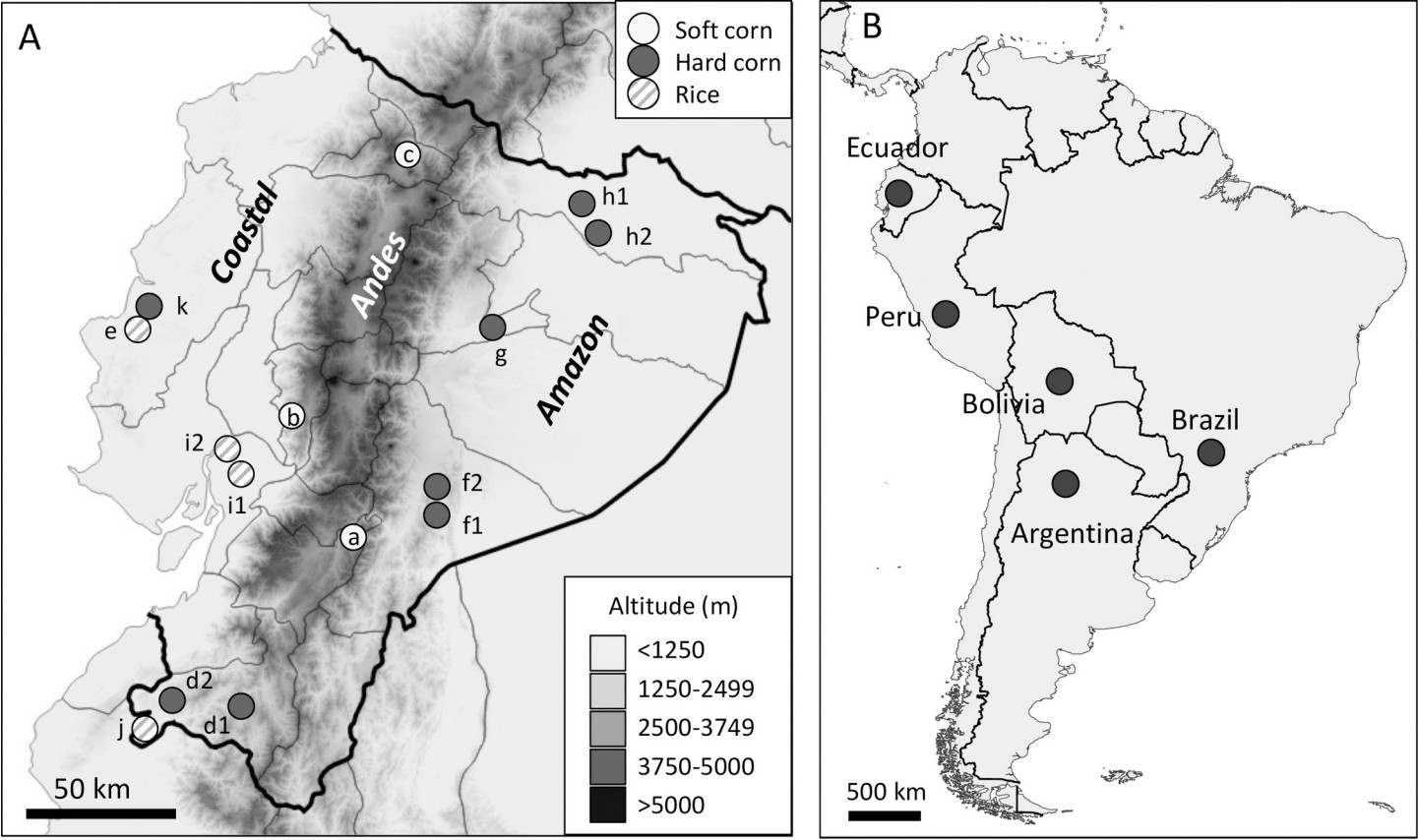

**Fig 1. Maps showing locations of fall armyworm collection sites as described in** [Table 1](). A: Map of Ecuador displaying outline of provinces and approximate locations of collection sites. The three major ecological regions (Coast, Andes, Amazon) are noted with differences in shading approximating elevation (meters above sea level). Circle patterns indicate the predominant host plant. Location letters refer to information from [Table 1](). B: Map of most of the northern half of South America showing locations of collection sites outside of Ecuador used in this study. The sites were in the vicinity of Lima (Peru), Mount Hope (Trinidad-Tobago), and Cotoca (Bolivia). Multiple provinces/states were surveyed in Brazil and Argentina with the most northern locations, Campo Verde (Brazil) and Salta (Argentina), identified on the map.

plates. All primers were obtained from Integrated DNA Technologies (Coralville, IA) and are mapped in [Fig 2](). Amplification of the COIB segment typically used the primer pair *924F* (5′ – TTATTGCTGTACCAACAGGT–3′) and *1303R* (5′ – CAGGATAGTCAGAATATCGACG–3′). Amplification of the TpiEI4 segment used the primers *412F* (5′ – CCGGACTGAAGGTTATC GCTTG –3′) and *1140R* (5′ – GCGGAAGCATTCGCTGACAACC–3′) to produce a variable length fragment due to insertion and deletion mutations in the intron portion. Nested PCR was used when needed with the first PCR done with primers *634F* (5′ –TTGCCCATGCTCTT GAGTCC–3′) and *1166R* (5′ –TGGATACGGACAGCGTTAGC–3′) and the second PCR using the internal primers *412F* and *1140R*.

For fragment isolations, 6 µl of 6X gel loading buffer was added to each amplification reaction and the entire sample run on a 1.8% agarose horizontal gel containing GelGreen (Biotium Inc, Fremont, CA) in 0.5X Tris-borate buffer (TBE, 45 mM Tris base, 45 mM boric acid, 1 mM EDTA pH 8.0). Fragments were visualized on a blue light illuminator and manually cut out from the gel. Fragment isolation was performed using Zymo-Spin I columns (Zymo Research, Orange, CA) according to manufacturer's instructions. Genewiz (South Plainfield, NJ) performed the DNA sequencing.

DNA alignments and consensus building were performed using MUSCLE (multiple sequence comparison by log-expectation), a public domain multiple alignment software

# A. *COI* gene (mitochondria)

# B. *Tpi* gene (Z-chromosome)

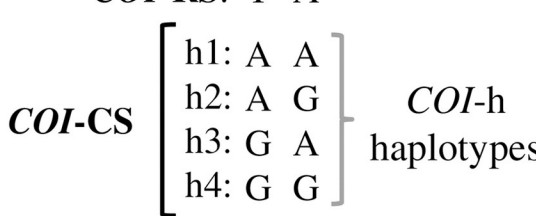

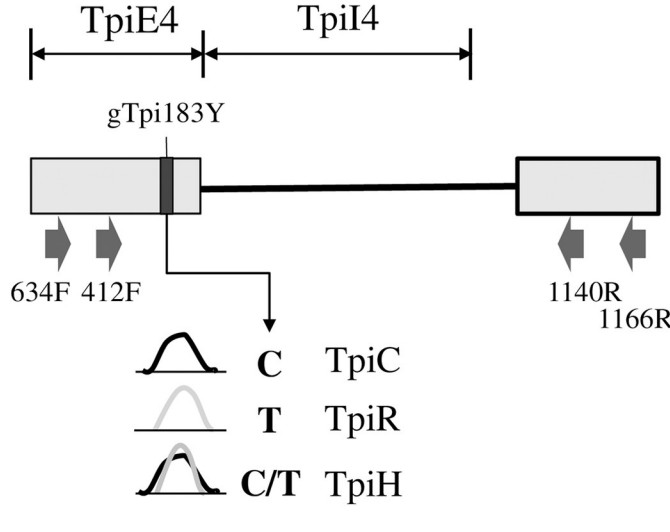

**Fig 2. Diagrams of the segments from the *COI* and *Tpi* genes used for the genetic analysis.** A: COIB gene segment identifying PCR primers used to amplify the fragment (block arrows) and the strain and COI-h haplotype defining the mCOI1164D and mCOI1287R polymorphic sites. *CO1*-RS is defined as a T and A at mCOI1164D and mCOI1287R, respectively. There are four corn-strain (*CO1*-CS) haplotypes (h1-h4) with and A or G observed at both mCOI1164D and mCOI1287R. B: Portion of the fall armyworm *Tpi* gene with block arrows indicating PCR primers. The TpiE4 exon segment contains the gTpi183 site that is polymorphic for a C or T. Representative DNA chromatograph patterns are shown to illustrate how TpiC, TpiR, and TpiH are defined. The TpiI4 intron segment lies adjacent.

incorporated into the Geneious Pro 10.1.2 program (Biomatters, New Zealand, http://www.geneious.com) [49]. Phylogenetic trees were graphically displayed in a neighbor-joining (NJ) tree analysis also included in the Geneious Pro 10.1.2 program [50].

## Characterization of the *COI* and *Tpi* gene segments

The genetic markers identifying strains are all single nucleotide substitutions (Fig 2). Sites in the *COI* gene are designated by an "m" (mitochondria) while *Tpi* sites are designated "g" (genomic). This is followed by the DNA name, number of base pairs from the predicted translational start site (*COI*) or 5' start of exon (*Tpi*) and the nucleotides observed using IUPAC convention (R: A or G, Y: C or T, D: A or G or T). The COIB sites mCOI1164D and mCOI1287R are diagnostic for strain identity in Western Hemisphere populations where there is a single rice-strain, $T_{1164}A_{1287}$, and four corn-strain configurations (COI-h haplotypes), $A_{1164}A_{1287}$ (h1), $A_{1164}G_{1287}$ (h2), $G_{1164}A_{1287}$ (h3), $G_{1164}G_{1287}$ (h4) (Fig 2A [51]).

Variants in the *Tpi* e4 exon segment (TpiE4) can also be used to identify host strain identity with results generally comparable with the *COI* marker[27]. The gTpi183Y site is on the fourth exon of the predicted *Tpi* coding region and was PCR amplified using the *Tpi* primers 412F and 1140R (Fig 2B). The C-strain allele (TpiC) is indicated by a $C_{183}$ and the R-strain (TpiR) by $T_{183}$ [27]. The *Tpi* gene is located on the Z-chromosome that is present in one copy in females and two copies in males. Because the genomic DNA was directly sequenced, males heterozygous for *Tpi* alleles will simultaneously display both alternatives at polymorphic sites,

which if different are easily identified by overlapping sequencing chromatographs. Heterozygosity at site gTpi183Y was limited to C/T and was denoted as TpiH.

The TpiI4 segment includes an approximately 172 bp portion of the adjacent intron, which is of variable length due to frequent insertions and deletions (indels). The segment was sequenced with primer 412F for the initial sequencing reaction and 1140R for 2nd strand sequence confirmation when needed in cases of ambiguity. The TpiI4 segment was chosen for analysis because it empirically had the most consistent sequence quality with the given primers. A variable but often substantial percentage of specimens were heterozygous for frameshift mutations in the intron that could be identified by overlapping chromatographs immediately following the polymorphism. These were not further analyzed.

## Statistical and data analyses

Quantification of genetic variability was made by calculations of haplotype diversity (Hd) and nucleotide diversity (Pi), which is a measure of the average number of nucleotide differences between randomly chosen sequences from a population. These were performed using the DNAsp software package [52]. Statistical analyses were conducted using GraphPad Prism version 7.00 for Mac (GraphPad Software, La Jolla California USA). Generation of graphs were done using Excel and Powerpoint (Microsoft, Redmond, WA). Geographical maps were generated using QGIS version 2.18.2 (Open Source Geospatial Foundation). Digital elevation data were downloaded from http://viewfinderpanoramas.org/dem3.html and processed by QGIS.

## Results

### Strain distribution between host plants

Our previous study analyzed fall armyworm collected from corn in the province of Manabi, Ecuador [48]. These data were compared to additional sites in Ecuador with collection numbers pooled by province and host plant for analysis using COIB polymorphisms (Fig 2A) to distinguish between the R-strain (*COI*-RS) and C-strain (*COI*-CS). Of the 492 specimens tested from Ecuador for COIB, 78 were from soft corn, 290 from hard corn, and 124 from rice. There were no differences observed between the host plants as 99% (488/492) expressed the *COI*-CS haplotype indicative of the C-strain (Fig 3A). The four *COI*-RS exceptions were found as single specimens from Azuay (a), Imbabura (c), Loja (d), and Sucumbíos (h) provinces, with none coming from the rice collections. The observations from Ecuador rice collections contrast with a previous survey from an Argentina rice habitat where 96% of the fall armyworm specimens expressed the *COI*-RS haplotype (Fig 3A).

Confirmation of the COIB-base strain identification was done by analysis of TpiE4 using 61 specimens from soft corn, 242 from hard corn, and 128 from rice. Because the *Tpi* gene is Z-linked and therefore present in two copies in fall armyworm males, the TpiE4 polymorphisms can identify interstrain hybrids as well as the two strains. Hybrids are designated TpiH and are associated with the presence of both polymorphic forms in the same specimen (Fig 2B). As with the COIB markers, nearly all the specimens collected from soft corn and hard corn expressed the C-strain TpiC configuration, with the only exceptions being seven TpiH hybrids that represent 4% of the previously examined collections from Manabi (Fig 3B). While the predominance of *COI*-CS and TpiC is expected for collections from corn, the frequencies of the R-strain markers were unusually low. For example, fall armyworm larval collections from corn plants in North America, Brazil, and Argentina routinely contain levels of the *COI*-RS and TpiR markers that while highly variable, average approximately 20% of the specimens tested [8, 27, 47].

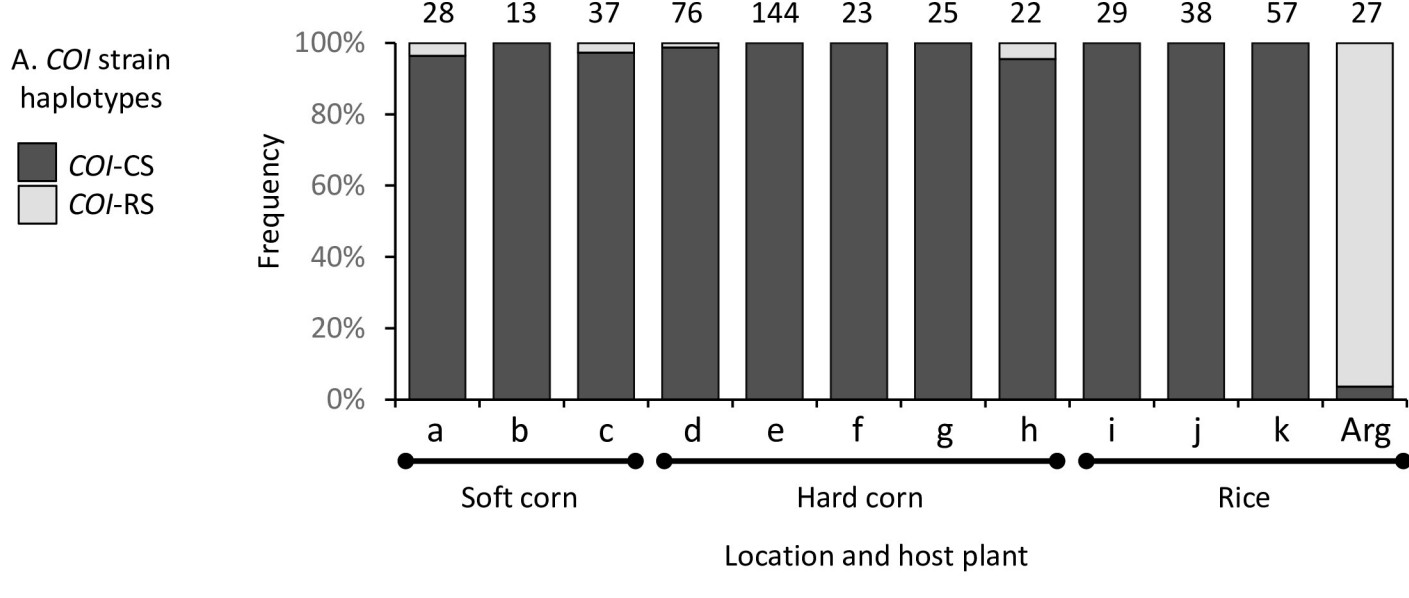

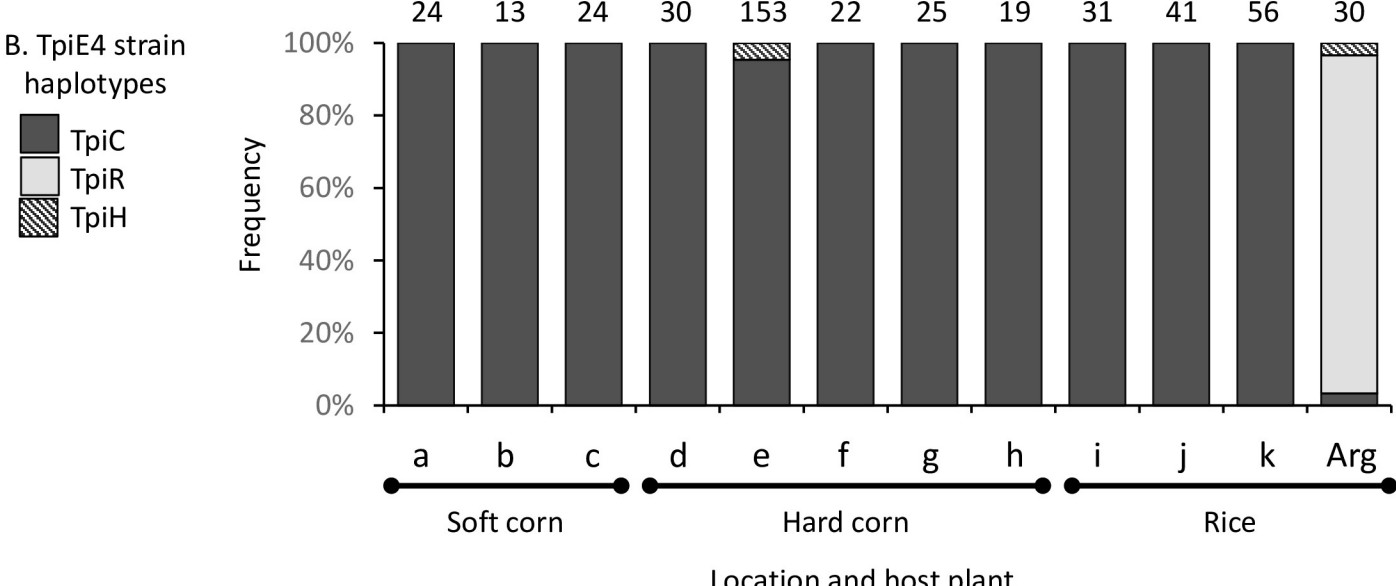

**Fig 3. Frequency distributions of the *COI* and *Tpi* strain haplotypes in different host plants and locations.** Fall armyworms collected from soft corn, hard corn, and rice in Ecuador were compared to collections from Argentina rice (Arg) describe in an earlier study [8]. A, Bar graph shows frequencies of *COI*-CS and *COI*-RS at locations described in Table 1. The *COI*-CS haplotype can be subdivided into four subgroups designated as the *COI*-h haplotypes. B, TpiE4 haplotype frequencies. Letters denote collection sites described in Table 1. Numbers above bars indicate the number of specimens tested.

Similar results were observed with the specimens from rice. Of the 128 larvae tested no TpiR were found, which contrasts with 93% TpiR frequency observed in the rice collection from Argentina (Fig 3B). The only evidence found for the presence of the TpiR haplotype comes from seven TpiH specimens collected from hard corn in the Manabi province, which made up 5% (7/153) of that collection.

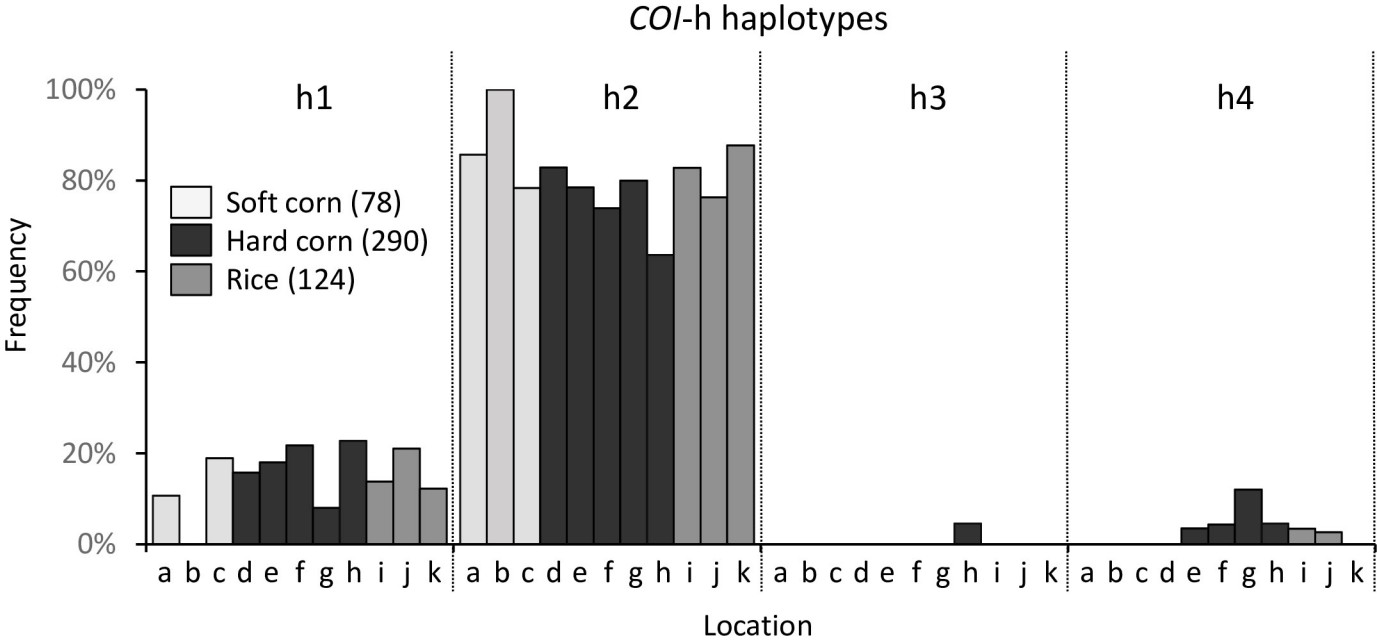

**Fig 4. Frequency distribution of the *COI*-h haplotypes relative to host plants and location.** The *COI*-CS group can be subdivided into four subgroups designated as the *COI*-h haplotypes. Letters on the x-axis denote provinces (from Table 1) and are organized by haplotype category. Numbers in parentheses indicate the number of specimens from each host plant.

## *COI*-h haplotype distribution

Polymorphisms in the COIB region subdivides the *COI*-CS group into the *COI*-h haplotypes (h1-h4, Fig 2A). The collections from the seven provinces are 99% *COI*-CS and these all show a similar *COI*-h haplotype profile regardless of host plant (Fig 4). The *COI*-h2 haplotype averaged 81% of all collections with a range from 64%-100%. The *COI*-h2 mean and standard deviation for soft corn (88 ± 11%), hard corn (75 ± 7%), and rice (81 ± 6%) were not significantly different by ordinary one-way ANOVA ($F = 2.2$, $P = 0.18$). The predominance of the *COI*-h2 haplotype is similar to that previously demonstrated for the Manabi province [48] and is characteristic of fall armyworm populations in Texas and the central United States, Mexico, and South America, but not Florida and the Caribbean [53].

## Characterization using TpiI4

The TpiI4 segment is a 172-bp sequence from a *Tpi* intron that was shown to be highly polymorphic and capable of distinguishing between the two strains (Fig 5A, [54, 55]). A total of 14 TpiI4 haplotypes were identified from 285 specimens from Ecuador and these were compared to 138 sequences composed of 84 haplotypes from five different countries in the Western Hemisphere. All 14 Ecuador sequences clustered with the C-strain haplotypes and showed no obvious bias to the haplotypes of a specific country (Fig 5B).

The relative frequencies of the 14 Ecuador TpiI4 haplotypes were calculated for the different collections (Fig 6). The Ecuador fall armyworm populations all showed a similar profile whether grouped by host plant (Fig 6A) or ecological regions (Fig 6B), with the Ecu02 haplotype was the most frequent and Ecu13 also consistently common. These two haplotypes were the major forms in other collections from South America (Fig 6C). The two closest surveyed populations to Ecuador (Peru and Bolivia) mostly displayed a subset of the Ecuador haplotypes with the one exception ("other") found in Peru. Only three TpiI4 haplotypes were found in

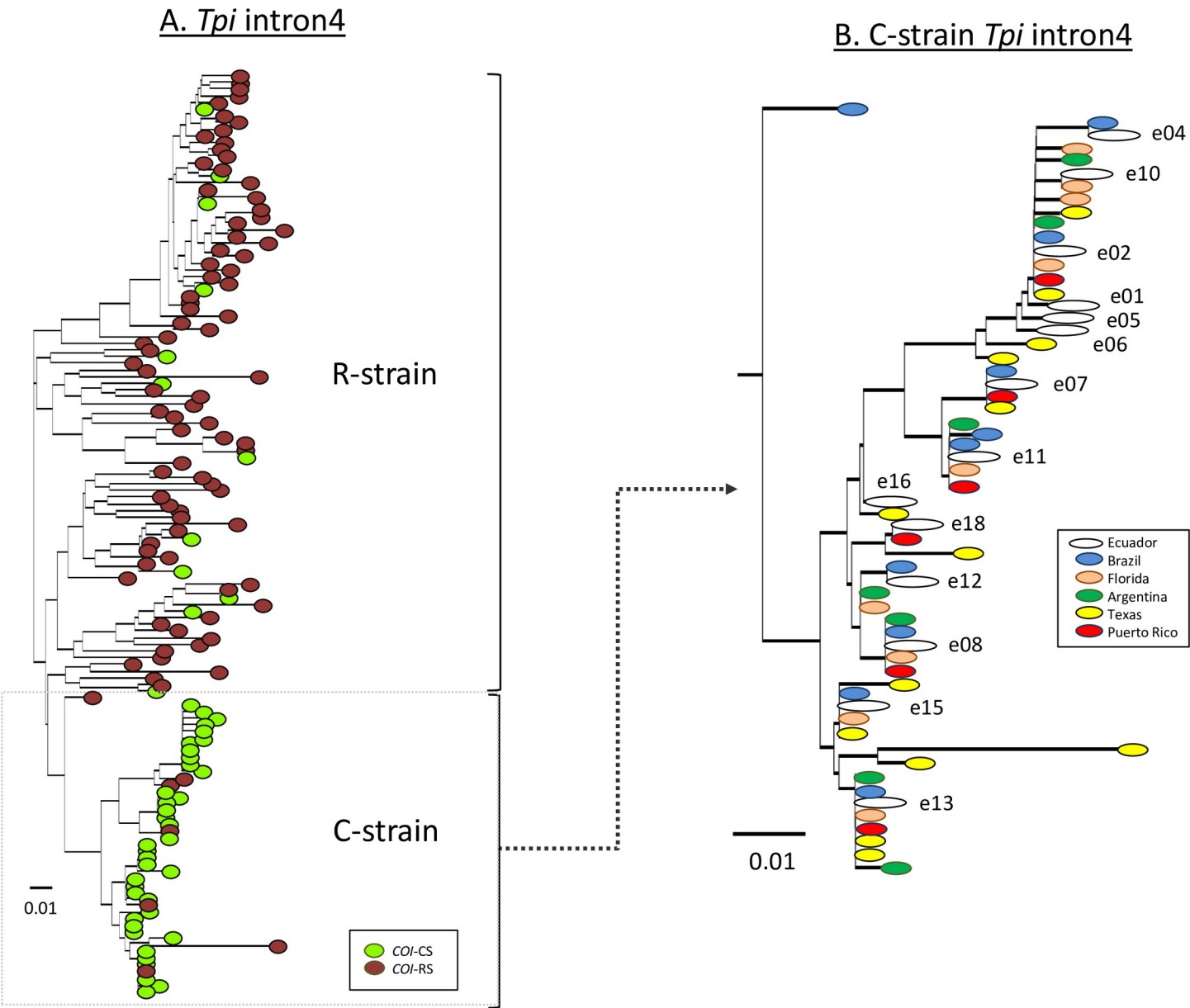

**Fig 5. Phylogenetic tree inferred by using the neighbor-joining method and Tamura-Nei genetic distance model [56].** The 14 Ecuador TpiI4 intron variants were compared to 84 unique haplotypes found from 138 Western Hemisphere larvae. A: The phylogenetic tree color-coded for the expression of *COI*-CS or *COI*-RS. B: The C-strain section of the phylogenetic tree where all 14 Ecuador sequences clustered is shown in expanded form and color-coded by location. Scale bar represents substitutions per site.

Bolivia, with Eco02 and Ecu13 predominant. Higher numbers of haplotype varieties were observed in the more southern collections in Brazil and Argentina, with 12% of Argentine haplotypes not observed in Ecuador (Fig 6C). Overall, 10 of the 14 Ecuador haplotypes were found in other locations. The four haplotypes unique to Ecuador (Ecu01, Ecu05, Ecu06, and Ecu15) were rare, combined they represent less than 4% of Ecuador fall armyworm.

To better quantify the similarities between the haplotype profiles we used Pearson *r* correlation analysis to assess the linear relationship between all locations and groupings. Significant

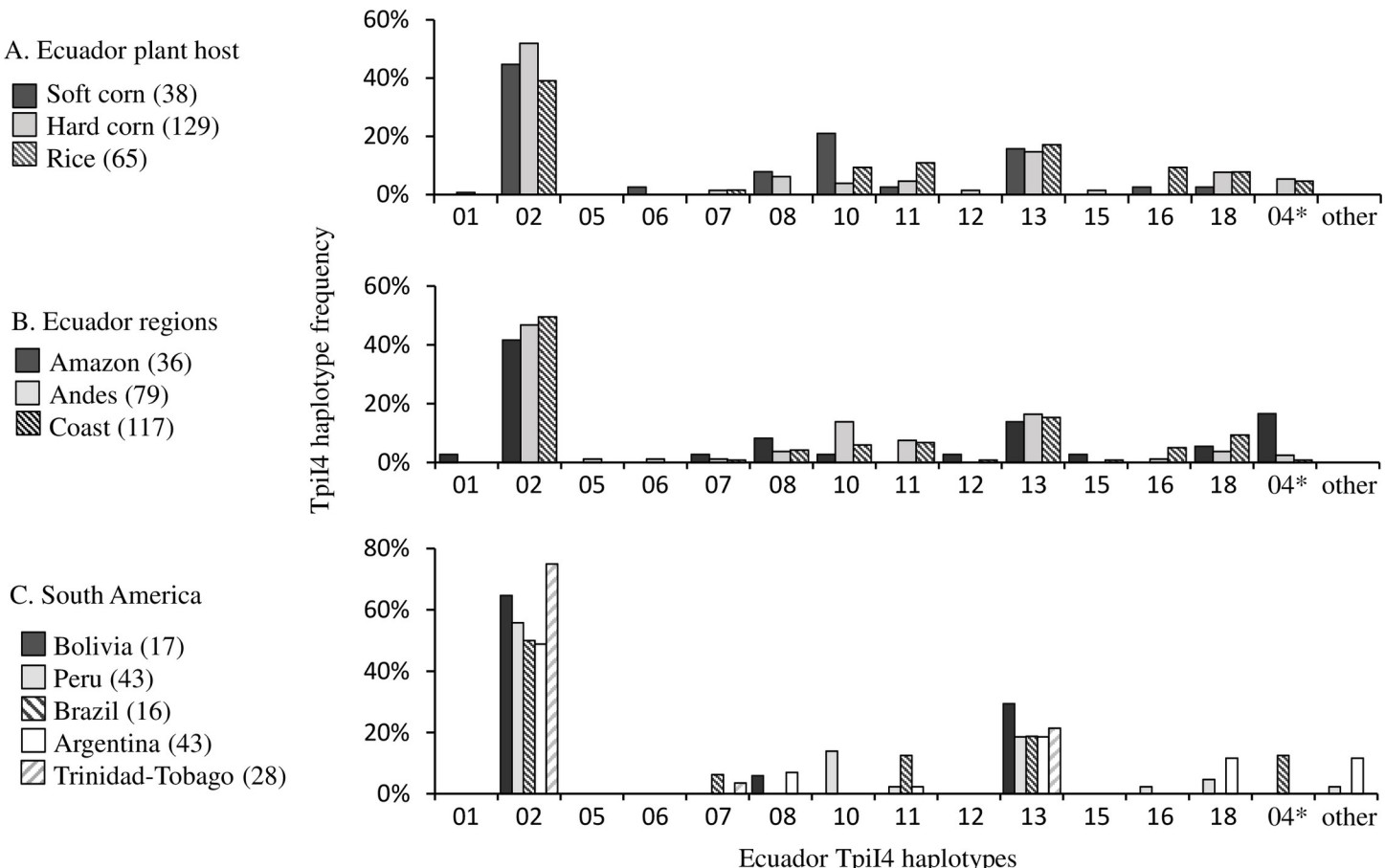

**Fig 6. Bar graph showing frequencies of the 14 TpiI4 haplotypes found in Ecuador (Ecu haplotypes) in different collections.** The number of sequences examined is in parentheses. Numbers on X-axis identify the TpiI4 Ecuador haplotypes. The Ecu04 haplotype (asterisk) is characterized by a 200-bp insertion that displays minor polymorphisms in sequence. The term "other" indicates haplotypes not found (to date) in Ecuador. A: Data from fall armyworm collected from soft corn, hard corn, and rice host plants in Ecuador. B: Data from collections grouped by ecological region. C: Data from South America (Peru, Bolivia, Argentina, Brazil) collections.

positive correlations were found for all pairwise combinations of the haplotype frequency profiles (Table 2), each indicative of a $P$-value < 0.001.

The genetic variability in the Ecuador populations as measured by heterozygosity (Hd = 0.736) is comparable to that observed in Brazil (0.811) and Argentina (0.718), higher than levels in Peru (0.651) and substantially higher than found in populations in Bolivia (0.503) or the Caribbean nation of Trinidad-Tobago (0.405, Table 3). Wright's $F_{st}$ statistic was calculated for the TpiI4 segment for all pair-wise combinations between Ecuador fall armyworm and the other collection sites. The $F_{st}$ values were low in all cases, ranging from negative numbers to a high of 0.0231 for the Ecuador/Trinidad-Tobago comparison. This indicates that most of the observed genetic variation is distributed within the populations, suggesting little genetic differentiation between Ecuador fall armyworm and those from the rest of South America and Trinidad-Tobago.

## Discussion

Isolated populations are expected to genetically diverge over time as measured by differences in the type and frequency of haplotypes. Differences in relative haplotype frequency are likely to be among the most sensitive indicators of restricted gene flow and has been shown to

**Table 2. Comparison of TpiI4 haplotype profiles from the different host plants and locations (described in Fig 3) as analyzed by Pearson (*r*) correlation tests.** Numbers represent Pearson *r* values. All pairwise comparisons showed a statistically significant positive correlation (*P* < 0.01). Abbreviated countries: Bolivia (Bol), Trinidad-Tobago (T&T), Brazil (Bra), and Argentina (Arg).

| | Soft corn | Hard corn | Rice | Amazon | Andes | Coast | Bol | Peru | Bra | Arg |
|---|---|---|---|---|---|---|---|---|---|---|
| Host plant | | | | | | | | | | |
| Hard corn | 0.91 | | | | | | | | | |
| Rice | 0.90 | 0.93 | | | | | | | | |
| Region | | | | | | | | | | |
| Amazon | 0.82 | 0.94 | 0.85 | | | | | | | |
| Andes | 0.97 | 0.96 | 0.96 | 0.88 | | | | | | |
| Coast | 0.92 | 0.99 | 0.97 | 0.90 | 0.97 | | | | | |
| South America | | | | | | | | | | |
| Bolivia | 0.90 | 0.97 | 0.91 | 0.91 | 0.94 | 0.96 | | | | |
| Peru | 0.97 | 0.96 | 0.95 | 0.88 | 0.99 | 0.98 | 0.96 | | | |
| Brazil | 0.82 | 0.94 | 0.92 | 0.92 | 0.92 | 0.92 | 0.93 | 0.90 | | |
| Argentina | 0.85 | 0.95 | 0.87 | 0.87 | 0.90 | 0.94 | 0.95 | 0.92 | 0.87 | |
| T&T | 0.89 | 0.98 | 0.91 | 0.91 | 0.95 | 0.97 | 0.98 | 0.97 | 0.94 | 0.94 |

consistently differentiate geographically separate fall armyworm populations that otherwise show the same haplotype composition [33, 46, 57].

Ecuador is subdivided into three regions, two of which (Coast and Amazon) are separated by the Andes mountain range that runs through the length of the country and has an average elevation of over 3,000 m (Fig 1A). This provides an opportunity to test the impact of the Andes on fall armyworm movements, which presumably involve a combination of natural and anthropogenic-assisted migration. The results show populations across Ecuador are homogeneous with respect to the distribution of the *COI*-h haplotypes (Fig 4) and analysis of the TpiI4 intron sequence reveals a statistically similar haplotype profile in all locations (Fig 6). These findings indicate that the Andes mountain range does not impose a barrier sufficient for significant divergence of haplotype types and frequencies, suggesting that fall armyworm is capable of long-distance flight at very high altitudes or that introgressions are occurring through human-assisted processes, such as the trade of contaminated products.

We previously presented evidence that the fall armyworm from cornfields in the Manabi province in Ecuador were genetically similar to those found in Peru, Bolivia, and Trinidad-Tobago, consistent with substantial mixing of these populations through migration [48]. We extended this comparison to Brazil and Argentina and continued to find general similarity in the *COI* and *Tpi* haplotype profiles that is consistent with the low $F_{st}$ values found in pairwise comparisons between these collections and those from Ecuador (Table 3). The lack of genetic

**Table 3. Genetic variability analysis for the TpiI4 segment for fall armyworm from different locations.** Metrics include number of sequences analyzed (n), number of haplotypes (N), haplotype diversity (Hd ± standard deviation [s.d.]), nucleotide diversity (Pi ± s.d.), and fixation index ($F_{st}$) measured in pair-wise comparison with Ecuador populations.

| Country | n | N | Hd ± s.d. | Pi ± s.d. | $F_{st}$ |
|---|---|---|---|---|---|
| Ecuador | 244 | 15 | 0.736 ± 0.026 | 0.029 ± 0.001 | - |
| Trinidad | 28 | 3 | 0.405 ± 0.094 | 0.021 ± 0.005 | 0.0231 |
| Peru | 43 | 7 | 0.646 ± 0.067 | 0.025 ± 0.004 | 0.0020 |
| Bolivia | 18 | 3 | 0.503 ± 0.103 | 0.027 ± 0.005 | -0.0166 |
| Brazil | 31 | 10 | 0.811 ± 0.055 | 0.029 ± 0.005 | -0.0173 |
| Argentina | 43 | 7 | 0.718 ± 0.059 | 0.031 ± 0.002 | 0.0112 |

differences between populations indicated by the $F_{st}$ analysis supports a common origin and recent interactions between at least the C-strain component of fall armyworm populations throughout South America.

Measures of genetic diversity commonly use the metrics of nucleotide diversity (Pi) and haplotype diversity (Hd) and we found that these provide some suggestions for how interactions are occurring between fall armyworm populations on the continent. The genetic variability as measured by Pi is consistent throughout the surveyed region, varying between a narrow range of 0.021 to 0.031 (Table 3). In contrast, haplotype diversity varies substantially, 0.405 to 0.811. These observations could be explained by weak population bottlenecks associated with migration from regions of high genetic diversity followed by the stochastic loss of rare haplotypes and the retention of more frequent variants. In this case, the number of polymorphic sites (nuclear diversity) is dictated primarily by the sequences of a shared set of the most common haplotypes and is therefore similar throughout the region, while haplotype diversity can vary substantially between migratory sources and destinations as haplotypes are lost. Under this scenario, locations with higher Hd (Ecuador, Argentina, and Brazil) are the likely source populations for migrations contributing to populations in Peru, Bolivia, and Trinidad-Tobago that exhibit less genetic diversity with a subset of shared haplotypes. Supporting the possibility that Ecuador can serve as a migratory source are extrapolations from climate suitability modeling that show favorable conditions in the country for permanent fall armyworm populations [48] and continuous farming of both corn and rice that provide consistent availability of high density host plant acreage.

In contrast with the C-strain, the R-strain appears to be very rare or absent in Ecuador, even in specimens collected from rice, a R-strain preferred host plant. The haplotype profiles of the fall armyworm capture from soft corn, hard corn, and rice in Ecuador are statistically indistinguishable from each other and significantly different from populations collected from R-strain hosts in other countries, such as Argentine rice (Fig 3). This to our knowledge is the first report of a geographical region where only a single fall armyworm strain is present independent of host plant. The extent of the region lacking the R-strain is uncertain. The R-strain has been reported in Colombia, which lies adjacent to Ecuador to the northeast [14, 15, 58] but specimens with R-strain markers have so far been absent in collections from Bolivia and Trinidad-Tobago, and only found rarely in Peru, leaving open the possibility that regions devoid of the R-strain may extend beyond Ecuador [59]. However, these latter collections were limited to corn host plants and so are not the ideal test for the presence of the R-strain. At this time, the presence of the R-strain based on the most rigorous definition, the biased distribution of *COI*-RS and TpiR on R-strain-preferred host plants, has been demonstrated in Argentina [8], Brazil [47], Colombia [58], and the southeastern United States [27, 60]. In addition, R-strain markers were found in substantial numbers in collections from several island nations in the Caribbean [59].

An alternative explanation for the near absence of R-strain markers in Ecuador is sampling bias, though we think this is unlikely. While the fall armyworm R-strain was originally discovered in rice there is evidence that this crop may not be a primary host for the R-strain specifically or fall armyworm in general. One study in Argentina showed variable strain specificity in rice collections [61], and in the United States the fall armyworm is only a minor and sporadic pest of rice. In comparison, the R-strain is consistently found in pasture and turf grass habitats where it predominates over the C-strain, indicating that these grass species are likely to be the principle preferred hosts for this population [9, 31, 60, 62]. Nevertheless, studies from multiple laboratories and locations indicate that when infestations in rice do occur, they usually have an R-strain majority [8, 15, 31, 47, 63, 64]. Therefore, we believe that the lack of any increase in R-strain markers in the Ecuador rice collections is strong evidence for the rarity of the R-strain in the sampled areas.

The infrequency, if not absence, of the R-strain in Ecuador indicates that there may be significant differences in the migratory behavior of the two strains in South America. One possible explanation for this comes from the modeling of fall armyworm migration in North America. Fall armyworm cannot survive the freezing winters characteristic of most of the United States and so the northern limit of major fall armyworm populations in the winter are located in southern Texas/northern Mexico and southern Florida [4]. As temperatures warm, fall armyworm from these locations move northward through the spring and summer in annual migrations that extend thousands of kilometers into Canada [38]. Studies combining modeling with haplotype mapping have shown that this migratory behavior can be explained by a combination of favorable wind patterns and the northward progression of corn planting. The latter provides the resources that can support high-density fall armyworm populations and highlights the relevance of host plant distribution for migration [38, 65]. It is possible that agricultural practices (*i.e.*, crop choice) and habitat distributions in at least parts of northern South America are supportive of C-strain but not R-strain migration, resulting in the observed regional differences in strain distribution.

Overall, these results have important implications for fall armyworm management in Ecuador. The predominance of only a single strain indicates that the host range of fall armyworm in Ecuador should be much less than observed in other locations. In particular, crop systems preferred by the R-strain such as pastures, forage grasses, and millet, should not be at risk for consistent and substantial infestation. However, the genetic similarity of the C-strain populations in South America suggest significant exchanges between these populations and substantial gene flow. This means that deleterious traits such as those that increase resistance to pesticides could spread rapidly into or out of Ecuador. A more detailed assessment of this risk will require additional studies to better map the patterns of migration in the region.

Fall armyworm has become a global pest with the potential to impose considerable economic damage to a wide range of crops. Assessing the risks of such infestations require knowledge of migration patterns and a better understanding of host strain differences that influence the distribution of fall armyworm populations. The observation of strain differences in the geographical distribution in parts of South America can be used to identify the environmental factors influencing the migration of this important agricultural pest, with applications to the mitigation of infestations and risk assessments. The presence of only a single strain in the Eastern Hemisphere and Ecuador has significant implications to assessments of what crops are at risk in these regions and underline the importance of strain identification and sampling of multiple crop types in the analysis of fall armyworm populations.

## Acknowledgments

We recognize Dr. J.M.G. Thomas for technical assistance in preparing the specimens and for comments about the manuscript. We thank technicians from INIAP, Verónica Quimbiamba, María Nieto and José Ullauri for supporting sampling logistics in Bolívar, Imbabura and Loja provinces, Ecuador. The use of trade, firm, or corporation names in this publication is for the information and convenience of the reader. Such use does not constitute an official endorsement or approval by the United States Department of Agriculture or the Agricultural Research Service of any product or service to the exclusion of others that may be suitable.

## Author Contributions

**Conceptualization:** Rodney N. Nagoshi, Sandra Garcés-Carrera.

**Data curation:** Rodney N. Nagoshi, Ernesto Cañarte, Bernardo Navarrete, Jimmy Pico, Catalina Bravo, Myriam Arias de López, Sandra Garcés-Carrera.

**Formal analysis:** Rodney N. Nagoshi.

**Funding acquisition:** Rodney N. Nagoshi, Sandra Garcés-Carrera.

**Investigation:** Rodney N. Nagoshi, Ernesto Cañarte, Bernardo Navarrete, Jimmy Pico, Catalina Bravo, Myriam Arias de López, Sandra Garcés-Carrera.

**Methodology:** Rodney N. Nagoshi.

**Project administration:** Rodney N. Nagoshi, Sandra Garcés-Carrera.

**Resources:** Rodney N. Nagoshi, Sandra Garcés-Carrera.

**Supervision:** Rodney N. Nagoshi, Sandra Garcés-Carrera.

**Validation:** Rodney N. Nagoshi.

**Visualization:** Rodney N. Nagoshi.

**Writing – original draft:** Rodney N. Nagoshi.

**Writing – review & editing:** Rodney N. Nagoshi, Ernesto Cañarte, Bernardo Navarrete, Jimmy Pico, Catalina Bravo, Myriam Arias de López, Sandra Garcés-Carrera.

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
