## [Decision Letter · Decision Letter 0]

15 Jun 2020

PONE-D-20-11128

The genetic characterization of fall armyworm populations in Ecuador and its implications to migration and pest management in the northern regions of South America.

PLOS ONE

Dear Dr. Nagoshi,

Thank you for submitting your manuscript to PLOS ONE. After careful consideration, we feel that it has merit but does not fully meet PLOS ONE’s publication criteria as it currently stands. Therefore, we invite you to submit a revised version of the manuscript that addresses the points raised during the review process.

We look forward to receiving your revised manuscript.

Kind regards,

Tzen-Yuh Chiang

Academic Editor

PLOS ONE

Journal Requirements:

Reviewers' comments:

Reviewer's Responses to Questions

**Comments to the Author**

1. Is the manuscript technically sound, and do the data support the conclusions?

Reviewer #1: Yes

2. Has the statistical analysis been performed appropriately and rigorously? 

Reviewer #1: Yes

3. Have the authors made all data underlying the findings in their manuscript fully available?

Reviewer #1: Yes

4. Is the manuscript presented in an intelligible fashion and written in standard English?

Reviewer #1: Yes

5. Review Comments to the Author

Reviewer #1: The manuscript by Nagoshi et al. (The genetic characterization of fall armyworm populations in Ecuador and its implications to migration and pest management in the northern regions of South America) expands on previous work by the authors and provides valuable information on fall armyworm population structure and apparent movement patterns in South America. The study seems to be well-executed, analyzed and interpreted, and the manuscript is well-written. I think the manuscript is acceptable for publication with minor revisions.

To help the reader understand the results and implications of this study, the authors should provide some additional detail on the fall armyworm collections. In particular:

• How were collection locations in Ecuador chosen? Do they correspond to major cropping areas? Are they separated by significant physical barriers?

• When were collections made (date)? Do they span some period of time or were they largely synchronous?

6. PLOS authors have the option to publish the peer review history of their article (what does this mean?). If published, this will include your full peer review and any attached files.

Reviewer #1: Yes: Graham Head

---

## [Author Response · Author response to Decision Letter 0]

2 Jul 2020

Reviewer comments in <italics>: 

< To help the reader understand the results and implications of this study, the authors should provide some additional detail on the fall armyworm collections. In particular:

• How were collection locations in Ecuador chosen? Do they correspond to major cropping areas? Are they separated by significant physical barriers?

The description of sampling in Ecuador was expanded to include how sites were chosen and now mentioning that they were at major cropping areas (lines 118-121). We now mention the Andes Mountains as a potential significant barrier (lines 103-106) and indicate the location of the Andes relative to the collection sites in Figure 1. 

We also reorganized the data to now directly address the possibility of the Andes acting as a physical barrier by grouping the collections by region. This is shown in Table 2 and Fig 6 and discussed in lines 371-381.

<• When were collections made (date)? Do they span some period of time or were they largely synchronous?

The timing of the collections is described in lines 121-125 and are listed in Table 1.

---

## [Editor Report · Decision Letter 1]

14 Jul 2020

The genetic characterization of fall armyworm populations in Ecuador and its implications to migration and pest management in the northern regions of South America.

PONE-D-20-11128R1

Dear Dr. Nagoshi,

We’re pleased to inform you that your manuscript has been judged scientifically suitable for publication and will be formally accepted for publication once it meets all outstanding technical requirements.

Kind regards,

Tzen-Yuh Chiang

Academic Editor

PLOS ONE
---

## [Editor Report · Acceptance letter]

20 Jul 2020

PONE-D-20-11128R1 

The genetic characterization of fall armyworm populations in Ecuador and its implications to migration and pest management in the northern regions of South America. 

Dear Dr. Nagoshi:

I'm pleased to inform you that your manuscript has been deemed suitable for publication in PLOS ONE. Congratulations! Your manuscript is now with our production department. 

Kind regards, 

on behalf of

Dr. Tzen-Yuh Chiang 

Academic Editor

PLOS ONE